# Theory of photoinduced ultrafast switching to a spin-orbital ordered hidden phase

Jiajun Li[1], Hugo U.R. Strand[2,3,4], Philipp Werner[4] & Martin Eckstein[1]

Photo-induced hidden phases are often observed in materials with intertwined orders. Understanding the formation of these non-thermal phases is challenging and requires a resolution of the cooperative interplay between different orders on the ultra-short timescale. In this work, we demonstrate that non-equilibrium photo-excitations can induce a state with spin-orbital orders entirely different from the equilibrium state in the three-quarter-filled two-band Hubbard model. We identify a general mechanism governing the transition to the hidden state, which relies on a non-thermal partial melting of the intertwined orders mediated by photoinduced charge excitations in the presence of strong spin-orbital exchange interactions. Our study theoretically confirms the crucial role played by orbital degrees of freedom in the light-induced dynamics of strongly correlated materials and it shows that the switching to hidden states can be controlled already on the fs timescale of the electron dynamics.

[1] Department of Physics, University Erlangen-Nürnberg, 91058 Erlangen, Germany. [2] Center for Computational Quantum Physics, Flatiron Institute, 162 Fifth Avenue, New York, NY 10010, USA. [3] Department of Quantum Matter Physics, University of Geneva, 24 Quai Ernest-Ansermet, 1211 Geneva 4, Switzerland. [4] Department of Physics, University of Fribourg, 1700 Fribourg, Switzerland. Correspondence and requests for materials should be addressed to J.L. (email: cong.li@fau.de)

Photo-induced phase transitions open the intriguing perspective of controlling complex materials on ultra-short timescales, with promising applications in information storage and processing[1–3]. An intense laser pulse can impulsively create charge excitations, and induce electronic processes which cannot be described in terms of a quasi-equilibrium scenario. This gives rise to rich physics in particular in Mott–Hubbard insulators, which include a large family of transition-metal oxides and chalcogenides. The charge gap in these materials prevents rapid thermalization to a featureless hot electron state, and the cooperative interplay of spin, orbital, and charge orders[4,5] allows for hidden phases, which can only be reached via an ultra-short laser excitation but not along thermal pathways[6–13].

The sub-picosecond electron dynamics can have a decisive effect even on the long-lived final states of a photo-induced system, as it determines the initial state for the subsequent evolution of one or several order parameters[1,6,14] in a multi-dimensional energy landscape. However, a microscopic understanding of the mechanisms which can initiate the transition to a hidden order on electronic timescales is often missing. Theoretical descriptions of the ultra-fast dynamics in solids have progressed in the weakly correlated regime, where mean-field and perturbative studies of various intertwined orders are possible[15–17]. For strongly correlated systems, extensive studies of the Hubbard model have provided insights into different aspects, such as charge relaxation and thermalization processes in one-band Mott insulators[18,19], the renormalization of bands by screening[20,21], and proposals for the laser control of magnetism[22,23]. While most of these studies involve single-orbital models, both strong correlations and multi-orbital degeneracy must be taken into account in order to resolve the cooperative dynamics of different orders in Mott insulators and explore the landscape of hidden states.

In this work, we investigate the non-thermal evolution of the intertwined spin-orbital-ordering and a resulting hidden phase in transition-metal compounds with a partially filled $d$-shell. In the representative case of one electron (or hole) in two $e_g$-orbitals, such as $d^4$ or $d^9$ configurations, spin and orbital exchange interactions can emerge due to the superexchange mechanism[24–26], and result in a spatially ordered pattern for both the spin orientations and orbital occupations[27,28]. The orbital-ordering drives the lattice to form Jahn–Teller-like distortions[29–31]. This scenario offers the intriguing opportunity to simultaneously switch spin and orbital orders through non-equilibrium protocols on the ultra-fast timescale. We demonstrate that in this situation, laser-induced charge excitations partially quench spin and orbital order on electronic timescales in a way that markedly differs from the effect of heating. As a consequence, the spin-orbital exchange drives the system to a transient hidden phase with a new orbital-order polarization on the picosecond timescale. The subsequent electron-lattice relaxation should lead to lattice distortions following the orbital-ordering on the timescale of picoseconds, which may be detected in experiments.

## Results

**Spin and orbital order in the two-band Hubbard model.** We consider a system with a partially filled $3d$ band in a cubic crystal, such that the $d$-shell is split into two $e_g$ and three $t_{2g}$ orbitals. Typical representatives are the perovskites, with a cubic arrangement of transition-metal ions in an octahedral environment of ligand atoms[32]. We assume the $t_{2g}$ orbitals are inactive (filled or empty), so that the system can be described by a two-band Hubbard model with two $e_g$ orbitals $d_{x^2-y^2}$ and $d_{3z^2-r^2}$ at

each site[24]. The local interaction is given by

$$H_U = U \sum_{i\ell} n_{i\ell\uparrow} n_{i\ell\downarrow} + \sum_{i,\sigma\sigma',\ell\neq\ell'} (U' - J_H \delta_{\sigma\sigma'}) n_{i\ell\sigma} n_{i\ell'\sigma'}$$
$$+ J_H \sum_{i,\ell\neq\ell'} \left( c^\dagger_{i\ell\uparrow} c^\dagger_{i\ell\downarrow} c_{i\ell'\downarrow} c_{i\ell'\uparrow} + c^\dagger_{i\ell\uparrow} c^\dagger_{i\ell'\downarrow} c_{i\ell\downarrow} c_{i\ell'\uparrow} \right), \quad (1)$$

where $i$ labels sites and $\ell = d_{x^2-y^2}, d_{3z^2-r^2}$ is the orbital index. $J_H$ is the Hund's coupling and $U' = U - 2J_H$. The hopping Hamiltonian is given by

$$H_0 = -t_0 \sum_{\langle ij\rangle \ell\ell'\sigma} e^{i\phi_{ij}(t)} c^\dagger_{i\ell\sigma} \hat{T}^\alpha_{\ell\ell'} c_{j\ell'\sigma}, \quad (2)$$

where the structure of the $2 \times 2$ hopping matrices $\hat{T}^\alpha$ along the bonds $\langle ij\rangle \parallel \alpha = x, y, z$ is imposed by the cubic symmetry, and electric fields can be included via a Peierls phase $\phi_{ij}$ (see methods). The hopping amplitude $t_0 = 1$ sets the energy scale. We use non-equilibrium dynamical mean-field theory (DMFT)[33] to solve this problem (see methods).

We consider the case of three-quarter-filling, and choose $U/t_0 = 7$ and $J_H/U = 0.1$ to roughly match the realistic parameter regime of KCuF3[31], with an insulating gap of $E_g \simeq W_{e_g} \approx 3$ eV. The time unit $\hbar/t_0$ and the initial temperature 0.01 then correspond to about 1 fs and 100 K, respectively. At this temperature, the DMFT calculation predicts A-type antiferromagnetic spin-ordering (A-AFM) and antiferro-orbital-ordering (AFO), consistent with both ab initio and mean-field results[29–31]. The local spins align ferromagnetically in the $xy$-plane and antiferromagnetically along the $z$-axis, and the hole approximately occupies the orbitals $d_{y^2-z^2}$ and $d_{x^2-z^2}$ in an alternating pattern (Fig. 1). To represent the orbital order, it is convenient to combine the two orbitals into a spinor $\hat{\psi}^T = \left( c_{x^2-y^2}, c_{3z^2-r^2} \right)$, and define the pseudospin components of the hole $Z_a = \frac{1}{2}\hat{\psi}\hat{\sigma}_a\hat{\psi}^\dagger$, with the Pauli matrices $\hat{\sigma}_{1,2,3}$. A transformation of the basis orbitals to $\left( d_{y^2-z^2}, d_{3x^2-r^2} \right)$ and $\left( d_{z^2-x^2}, d_{3y^2-r^2} \right)$ corresponds to successive $\theta = 120°$ rotations around the $Z_2$-axis, using the rotation matrix $\hat{R}(\theta) = e^{i\hat{\sigma}_2\theta/2}$, and the resulting pseudospin components in this new basis will be denoted by $X_a$ and $Y_a$, respectively (Fig. 1).

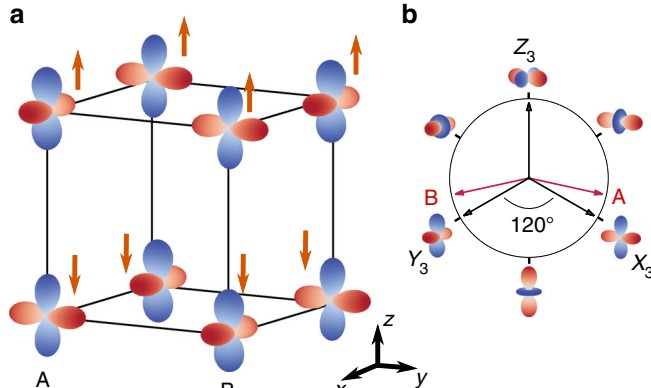

**Fig. 1** Spin- and orbital-ordered phase in equilibrium. **a** The spin-orbital order in a cubic environment in equilibrium. Each site contains three electrons and the unoccupied orbital is shown, with arrows labelling the total spin moment of the electrons. The color scale indicates the value of orbital spherical harmonics in arbitrary units. **b** The $Z_1$-$Z_3$ plane in the orbital pseudospin space (the compass). The $X_3$, $Y_3$, $Z_3$ directions and their corresponding orbitals are marked on the compass. The hole-occupied orbitals on A and B lattice sites are shown as red arrows

Concentrating on the site A in Fig. 1 and defining $d_{y^2-z^2}$ and $d_{3x^2-r^2}$ as orbital 1 and 2, respectively, the A site has two electrons in orbital 2 and one spin-down electron in orbital 1. The orbital-order parameter can be defined as the occupation difference between the two orbitals, which is the component $X_3 = \frac{1}{2}(n_2 - n_1)$ of the orbital pseudospin. The orbital pseudospin vector $\langle \mathbf{X}_B \rangle$ on the B sites is a reflection of $\langle \mathbf{X}_A \rangle$ w.r.t. the $Z_3$-direction, i.e., the order parameter approximately alternates between the $X_3$ and $Y_3$ directions on neighboring lattice sites, and the ordered phase is invariant under the transformation $\hat{\psi}_{i\sigma} \rightarrow \hat{\sigma}_3 \hat{\psi}_{i+\hat{e}_{x/y},\sigma}$ and $\hat{\psi}_{i\sigma} \rightarrow \hat{\sigma}_3 \hat{\psi}_{i+\hat{e}_z,-\sigma}$. This symmetry is preserved out of equilibrium, so that the general spin-orbital order is given by a three-dimensional manifold parametrized by the magnitudes of $S_z$ and $X_3$ and the direction of the pseudospin, $\tan \theta = X_3/X_1$.

**Photo-induced reduction of spin and orbital order**. We consider three non-equilibrium protocols: excitation of the $e_g$ system with an electric field pulse with polarization parallel or perpendicular to the ferromagnetic planes, and a photo-doping, i.e., a sudden excitation of electrons into (out of) the $e_g$ manifold by a resonant laser excitation from (to) other bands (see methods). We first study the relaxation after an electric pulse polarized in the diagonal of the $xy$-plane, taken to be a single cycle pulse of period $T \approx 2$. The pulse creates non-equilibrium charge excitations and causes a reduction or melting of the spin and orbital orders, as shown in Fig. 2. Furthermore, while the equilibrium orbital pseudospin $\mathbf{X}$ corresponds to a real superposition of orbital states in the $X_1 - X_3$ plane, transiently a small non-zero $X_2$ component emerges, indicating precession dynamics induced by the excitation[34]. The long-time relaxation of order parameters can be

examined by fitting the time evolution of both $S_z$ and $X_3$ by exponential functions, $S_z(t) = S_z(\infty) + C_S e^{-t/\tau_S}$ and $X_3(t) = X_3(\infty) + C_X e^{-t/\tau_X}$, to extract the decay rates $\tau_{X,S}^{-1}$ (Fig. 2c) and the extrapolated order parameters at $t = \infty$ (Fig. 2b). We analyze the relaxation as a function of the excitation density $n_{ex}$ (the photo-induced change of 4- and 2-electron configurations), by varying the amplitude of the pulse. The rate $\tau_S^{-1}$ falls below $\tau_X^{-1}$ as $n_{ex}$ increases, in line with the slower melting of spin order shown in Fig. 2a. Furthermore, in the long-time limit the AFM order $S_z(\infty)$ is systematically stronger than the AFO order $X_3(\infty)$, which eventually melts beyond a threshold $n_{ex} \simeq 0.015$. At the same threshold we observe a slow down of the melting, but no divergence of relaxation times as obtained for a second-order phase transition[35,36]. The slower and weaker melting of A-AFM order compared to the AFO order is also observed for a z-polarized excitation, and after photo-doping holes or electrons into the system. The observed behavior is exactly opposite to the thermal melting of spin order, which would precede the melting of orbital order because of a smaller spin exchange interaction[29].

**Hidden state**. In the long-time limit, the presence of charge excitations leads to a quasi-steady photo-excited state, which does not thermalize on the 100 fs timescale of the simulation due to the Mott gap. We now compare the multi-dimensional order parameter (given by $S_z(\infty)$, $X_3(\infty)$, and the angle $\tan \theta = X_1(\infty)/X_3(\infty)$) in the photo-excited state at different excitation densities, and in various equilibrium states. We first look at the relative magnitude of $X_3$ to $S_z$, by plotting $X_3$ against $S_z$ in different states (Fig. 3a). In thermal equilibrium, as the temperature increases, the $S_z - X_3$ curve first drops to $S_z = 0$ and then proceeds to the high-temperature state $X_3 = S_z = 0$, reflecting the lower critical temperature of the A-AFM order compared to the AFO order. Chemically doped systems follow similar paths, as shown by the red and blue dot-dashed lines for doping $\delta n = \pm 0.01$. The photo-excited states, in contrast, follow a smooth curve in the $S_z - X_3$ plane with $S_z > X_3$ when $n_{ex}$ is increasing, for all three non-equilibirum protocols. Photo-doping electrons leads to the weakest AFM order, indicating that 4-particle excitations most efficiently destroy the spin order. The photo-excited states at a given density $n_{ex} = 0.01$ (see square symbols) exhibit different order parameters than the equilibrium system with the same hole or electron doping $\delta n = \pm 0.01$, independent of temperature. Hence the states reached by ultra-fast laser excitation are not accessible under equilibrium conditions. Furthermore, the direction of $\mathbf{X}(\infty)$ in the pseudospin space shows that the polarization of orbital order is different in the equilibrium and non-equilibrium states (Fig. 3b). In equilibrium, the angle between $\mathbf{X}_A$ and $\mathbf{X}_B$ of the two sublattices (~120°) increases with temperature and stays at 180° after the AFM order has melted. On the contrary, photo-excitation results in a decreasing angle, which evolves towards 0° in the strong excitation limit, corresponding to a ferro-orbital (FO) ordering with a small magnitude $|\mathbf{X}|$.

We now explain the mechanism which drives the system into the hidden state. It follows from the non-thermal nature of quenching the orders by photo-induced carriers, and the spin-orbital exchange interactions, which act differently in this unconventional quenched state compared to the equilibrium state.

**Femtosecond quench of spin and orbital order**. While thermal melting of spin and orbital order is due to the population of (orbital) spin-waves, the partial quench of the two order parameters after photo-excitation follows an entirely different mechanism: It occurs on the femtosecond timescale, as the motion of charge excitations leaves a string of defects in the

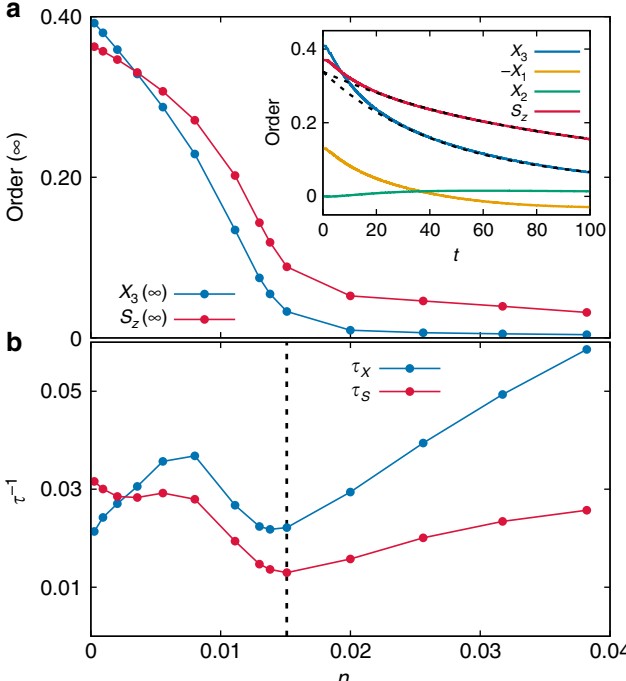

**Fig. 2** The photo-induced reduction of spin-orbital-ordering. **a** The extrapolated order parameters $X_3(\infty)$ and $S_z(\infty)$. Inset: Time evolution of spin and orbital-order parameters after the electric field pulse with $n_{ex} = 0.015$. Dashed lines show an exponential fit. **b** The fitted relaxation rate $\tau_S^{-1}$ and $\tau_X^{-1}$ for the spin ($S_z$) and orbital ($X_3$) order parameters, respectively. As the excitation density $n_{ex}$ increases, the spin relaxation rate drops below the rate for the orbital order. Both spin and orbital relaxations slow down close to $n_{ex} \sim 0.015$ (labelled by the vertical dashed line)

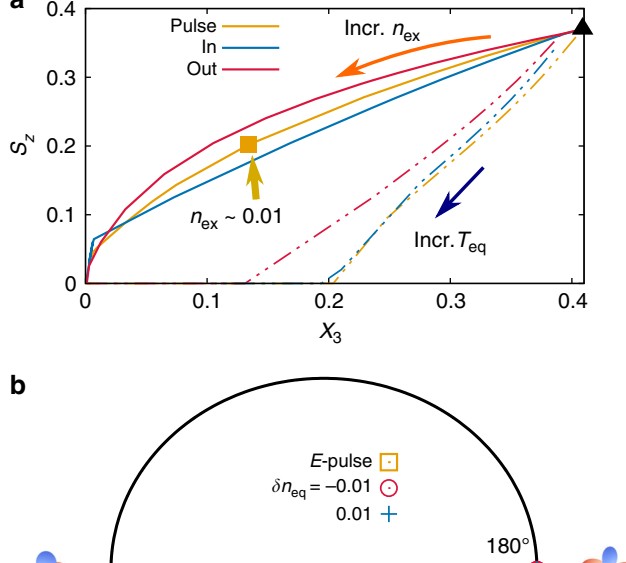

**Fig. 3** The extrapolated spin and orbital order in the long-time limit. **a** The three different non-equilibrium protocols (electric field pulse, photo-doping in/out electrons) all lead to stronger spin-ordering than orbital-ordering, which is qualitatively different from the behavior of the equilibrium system with increasing temperature (yellow double-dotted line) at integer filling and for the chemically doped systems with $\delta n = \pm 0.01$ (red/blue double-dotted lines). **b** Normalized orbital-order components $X_1$ and $X_3$ in the long-time limit shown on the pseudospin compass. Rising equilibrium temperature and photo-excitation cause the orbital pseudospin to rotate in opposite directions

ordered background, similar to the case of the single-band AFM[19,36,37]. Due to the in-plane ferromagnetism in the $xy$-plane, only the motion of charge excitations along the $z$ direction affects the spin order (process 1 in Fig. 4b), while orbital order can be affected by hopping processes in all three directions (process 2 in Fig. 4b) and thus should decrease faster.

To confirm this mechanism, we first note that it corresponds to a transfer of kinetic energy from the charge carriers to the ordered background. This can be seen directly by looking at the electronic distribution functions, which show the relaxation to a relatively cold distribution within $t \lesssim 50$ inverse hoppings (Fig. 4a). Furthermore, at early times, the anisotropy is also reflected in the polarization dependence of the excitation (Fig. 4c). During the pulse, the suppression of the magnetic order is lower for $E \parallel x, y$, consistent with the fact that this creates mostly in-plane spin-triplet excitations (process 4 in Fig. 4b), while the perpendicular polarization $E \parallel z$ directly creates spin-singlet doublons (process 3 in Fig. 4b), affecting A-AFM and AFO in the same manner. After several hopping times, the decay rates of the A-AFM and AFO order for both polarizations differ by roughly a factor of two to three, consistent with an independent melting of the two orders due to the hoppings along the different directions. Note that a larger Hund's coupling $J_H$ might further suppress out-of-

plane hopping by energetically penalizing the conversion of spin-triplet into spin-singlet doublons.

**Orbital-order polarization.** After the non-thermal reduction of $X_3$ and $S_z$, the unconventional polarization of the orbital order qualitatively follows from the intertwined dynamics of the two orders due to the spin-orbital exchange interactions. In the Mott phase at $U \gg t_0$, the latter are described by the Kugel–Khomskii model[29],

$$H = \sum_{\langle ij \rangle \parallel \hat{x}} \xi_x X_{3i} X_{3j} + \sum_{\langle ij \rangle \parallel \hat{y}} \xi_y Y_{3i} Y_{3j} + \sum_{\langle ij \rangle \parallel \hat{z}} \xi_z Z_{3i} Z_{3j} + \sum_i \left( \eta_x X_{3i} + \eta_y Y_{3i} + \eta_z Z_{3i} \right). \tag{3}$$

Here $\xi_\alpha, \eta_\alpha$ are orbital exchange parameters that depend on the spin configuration on the bond $\langle ij \rangle \parallel \alpha$ with $\alpha = x, y, z$. In particular,

$$\eta_\alpha = \frac{J}{2} \left[ \frac{1}{4} - \vec{S}^i \vec{S}^j \right], \tag{4}$$

with a positive exchange interaction $J$ obtained through the Schrieffer–Wolff transformation[38]. The parameter $\eta_\alpha$ is maximized for spin-singlet and minimized for spin-triplet, thus $\eta_z > \eta_x = \eta_y$ under A-AF spin order.

As $S_z$ vanishes for increasing temperatures, the compass parameters $\xi_\alpha$ and $\eta_\alpha$ become isotropic for $\alpha = x, y, z$, and the 180° orbital-ordering minimizes the mean-field energy[24,39]. In the photo-excited states, however, the compass parameters remain anisotropic with a finite $S_z$, while the strong reduction in $|\mathbf{X}|$ renders the linear terms on the second line of Eq. (3) dominant. Therefore, the pseudospins align along the negative $Z_3$-direction (orbital $d_{3z^2-r^2}$) due to the dominant $\eta_z$.

## Discussion

In summary, our finding suggests a pathway to reach hidden states in correlated electron systems with intertwined spin and orbital order on the ultimately short timescale of the electronic hopping. The non-thermal quench transfers energy from photo-induced charge excitations into the A-AFM and AFO ordered backgrounds at different rates. Due to the exchange-coupling between the order parameters, this drives the system to a state that features spin-orbital orders unaccessible in an equilibrium state. In particular, starting with a near-120° AFO ordering, the photo-excited system approaches a ferro-orbital-ordering in the strong excitation limit, while an equilibrium state (doped or not) always reaches a 180° AFO ordering with increasing temperature. An obvious target material to look for these effects is KCuF3, whose A-AFM and AFO orders are well described by the two-band Hubbard model. However, the general finding, i.e., that a photo-induced quench of a multi-component exchange-coupled order parameter can lead to hidden states on electronic timescales, should apply to a broader class of materials with other spin-orbital orderings, e.g., the manganites[5,26]. Such non-thermal electronic states are important as they initiate the subsequent dynamics of non-thermal order parameters[14]. In particular, the Jahn–Teller effect can be non-negligible in realistic perovskites[40,41], but the electronic mechanism would still be a key driving force, among other effects, of the full electron-lattice dynamics. One possible scenario is suggested by recent experiments[7,14], where the subsequent lattice dynamics is driven by the fast change in electronic degrees of freedom. This should be the case when the electron-lattice coupling is weak enough and only affects the dynamics on longer timescales than the electronic processes. In the strong coupling limit, on the other hand, the electrons can be dressed with lattice distortions to form

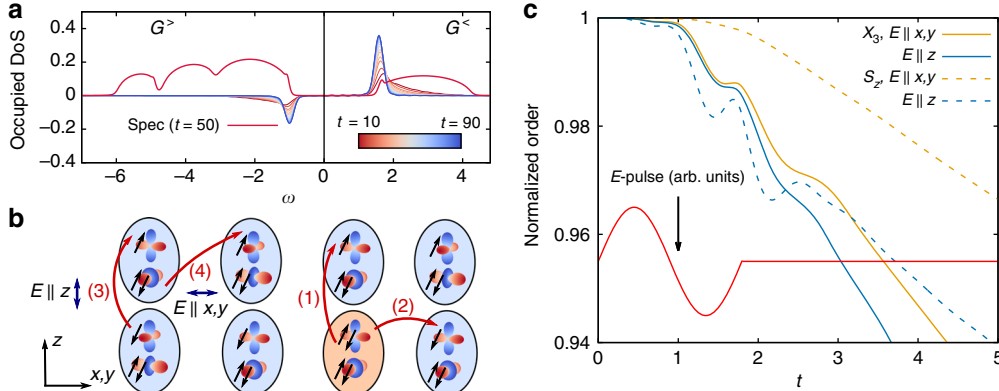

**Fig. 4** The electronic processes occuring in the photo-excited state. **a** Relaxation of charge excitations and the photo-excited state in the long-time limit. $G^<(\omega, t)$ at $\omega > 0$ corresponds to the distribution of 4-particle excitations at $\omega > 0$ while $G^>(\omega, t)$ at $\omega < 0$ shows the distribution of doublons (see methods). The red solid line is the density of states (spectral function) at $t = 50$. **b** The charge excitations created by pulses perpendicular/parallel to the ferromagnetic plane. A parallel pulse ($E \parallel x, y$) creates spin-triplet excitations in process 4, while a perpendicular pulse ($E \parallel z$) is polarized along the AF-axis and directly creates spin-singlet excitations in process 3. Processes 1 and 2 indicate possible hopping processes of 4-particle excitations. The lattice site of the 4-particle excitation is of orange color. Blue arrows indicate the polarization of the electric field. **c** The non-equilibrium melting of spin and orbital order under a one-cycle electric field pulse of $E_0 = 0.6$. The solid (dashed) lines show $X_3$ ($S_z$) normalized by their initial values

polarons. With renormalized parameters, the electronic mechanism of weaker and slower melting of AFO than A-AFM could still be established in the polaron dynamics. Thus, in both cases, the opposite rotation of the orbital pseudospin in the equilibrium and non-equilibrium regimes might reveal itself by inducing opposite Jahn–Teller-like distortions through electron-phonon coupling, which can possibly be detected by time-resolved X-ray diffraction techniques. The competition between Jahn–Teller effect and the electronic mechanism in the intermediate coupling regime can be more complicated and requires further studies.

The existence of new types of order upon photo-excitation is in sharp contrast to the one-band Hubbard model, where the excitation density and effective temperature exclusively determine the spectral properties and exchange interactions in the photo-excited state[36,42]. In addition to the multi-component order-parameter, the $e_g$-orbital degeneracy allows for multiple types of charge excitations (in particular, there are three different doubly occupied sites). Thus, even locally there are electronically excited states described by a continuum of parameters, potentially giving rise to many near-degenerate phases. In fact, in the present case, along with the different orders, also the probability distribution of the electronic excitations in the two-particle sector is found to be different from the equilibrium state. The multiple flavors of charge excitations may offer rich possibilities to engineer the exchange interactions[42], and thereby further control the dynamics on the ps timescale. It is worth noting that, although one can explicitly impose the ferro-orbital-ordering in a constrained DMFT simulation, the obtained solution is in general unstable in equilibrium. Hence, the presence of charge excitations should play a critical role in stabilizing the hidden phase. To investigate those possibilities in detail requires a significant extension of the simulation time. In future works, also a steady-state formalism might be adopted to directly study the electronic quasi-steady state and identify the general non-equilibrium protocols that allow to explore the hidden manifold of non-thermal phases.

## Methods

**The two-band Hubbard model.** The hopping matrices $\hat{T}^\alpha$ are imposed by the cubic symmetry, or in particular, the permutation symmetry of $x$, $y$, $z$–bonds. $\hat{T}^z$, the hopping along the $z$–bond, is determined by the only non-vanishing matrix element between $d_{3z^2-r^2}$ orbitals[29]. All other matrices can be determined through

rotations:

$$
\begin{aligned}
\hat{T}^z &= \begin{pmatrix} 0 & 0 \\ 0 & 1 \end{pmatrix}, \\
\hat{T}^x &= \hat{R}(\tfrac{4}{3}\pi)\hat{T}^z\hat{R}(-\tfrac{4}{3}\pi) = \tfrac{1}{4}\begin{pmatrix} 3 & -\sqrt{3} \\ -\sqrt{3} & 1 \end{pmatrix}, \\
\hat{T}^y &= \hat{R}(-\tfrac{4}{3}\pi)\hat{T}^z\hat{R}(\tfrac{4}{3}\pi) = \tfrac{1}{4}\begin{pmatrix} 3 & \sqrt{3} \\ \sqrt{3} & 1 \end{pmatrix}.
\end{aligned}
\tag{5}
$$

In the one-hole- (three-electrons) or one-electron-filled case, using a Schrieffer–Wolff transformation, an effective Hamiltonian (Kugel–Khomskii model[29]) in terms of spin and orbital pseudospin can be obtained, which preserves the threefold rotational symmetry in the pseudospin space. This model is an example of a compass model and has been intensively studied in the literature[25].

**Dynamical mean-field theory.** In non-equilibrium dynamical mean-field theory, the lattice system is approximated by an effective impurity model coupled to a non-interacting bath. The impurity action approximating the lattice problem takes the form $\mathcal{S} = \mathcal{S}_{loc} - i\sum_\sigma \int dt \int dt' \hat{\psi}_\sigma^\dagger(t')\hat{\Delta}^\sigma(t, t')\hat{\psi}_\sigma(t')$[33]. The two-band Hubbard model involves orbital-mixing terms, but conserves the total spin $S_z$ component. Therefore, the hybridization function of the bath $\hat{\Delta}_{\ell\ell'}^\sigma(t, t')$, as well as the Green's functions, are diagonal in the spin indices. We choose the non-crossing approximation (NCA) as the impurity solver[43], which yields reliable results when the two-band Hubbard model at large $U$ is considered[44].

The lattice consists of two sublattices A and B with different orbital occupations. The self-consistency condition for the hybridization function reads $\hat{\Delta}_A^\sigma(t, t') = \tfrac{1}{6}\sum_{\alpha,\zeta} e^{i\zeta\phi_\alpha(t)}\hat{T}^{\alpha\dagger}\hat{G}_{B,\alpha}^\sigma(t, t')\hat{T}^\alpha e^{-i\zeta\phi_\alpha(t)}$, where $\zeta = \pm 1$ corresponds to positive/negative directions along the same bond. This self-consistency represents a Bethe lattice in which $d$ bonds are connected to each lattice site[45] along each direction $\pm x$, $\pm y$, $\pm z$, and we take the limit of $d \to \infty$ with a rescaled hopping parameter $t_0/\sqrt{6d}$. This model has a (single-orbital) semi-circular density of states with bandwidth $W = 2t_0$, and one expects the results to be qualitatively similar to a cubic lattice of the same bandwidth.

Motivated by the mean-field solutions, we consider an intertwined spin and orbital-ordered phase, where A-type antiferromagnetism and antiferro-orbital order are assumed[24]. The orbital orders on the two sublattices can be related by a unitary rotation $\mathcal{R} = \hat{\sigma}_3$ in the pseudospin space which flips the pseudospin w.r.t. the $Z_3$-direction, i.e., $\hat{G}_{B,\alpha}^\sigma = \mathcal{R}^\dagger \hat{G}_{A,\alpha}^{\sigma'}\mathcal{R}$, where the spin $\sigma'$ is determined by the bond, with $\sigma' = \sigma$ for $\alpha = x, y$ and $\sigma' = -\sigma$ for $\alpha = z$.

**Characterization of the photo-excited state.** The laser excitation is included in the model by the Peierls phase, as indicated in Eq. (2). Photo-doping is realized by connecting the lattice to an empty or filled Fermion bath for a short time ($t \leq 2$). The photo-doped system can be characterized by the excitation density $n_{ex}$. $n_{ex}$ is defined as the sum of excited doublon and 4-particle excitations. In the electric-pulse case, this quantity can be calculated as the pulse-induced growth in the

probabilities of local 4-particle and 2-particle states. In the photo-doping case, it can be measured by the change in the total particle number after the photo-doping (coupling to the fermion bath).

In Fig. 4, the time-dependent density of states and occupied density of states are computed through Fourier transforms over the relative time variables $A(\omega, t) = -\mathrm{Im}\left\{\mathrm{Tr}_{\ell\sigma}\int ds e^{i\omega s}\widehat{G}^r(t+s/2, t-s/2)\right\}/\pi$ and

$G^{\lessgtr}(\omega, t) = \mathrm{Im}\left\{\mathrm{Tr}_{\ell\sigma}\int ds e^{i\omega s}\widehat{G}^{\lessgtr}(t+s/2, t-s/2)\right\}$, which are traced over orbitals and spins.

## Data availability

The data that support the findings of this study are available from the corresponding author upon reasonable request.

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

## Acknowledgements

We acknowledge the financial support from the ERC starting Grant No. 716648. The Flatiron Institute is a division of the Simons Foundation.

## Author contributions

M.E. and J.L. conceived the project. J.L. has run the DMFT simulations. M.E., H.U.R.S., and J.L. contributed to the non-equilibrium DMFT code. All authors contributed to the discussion and the writing of the manuscript.

## Additional information

**Competing interests:** The authors declare no competing interests.

