## [Peer Review File · Nature Communications]

Reviewers' Comments:

Reviewer #1:

Remarks to the Author:

The manuscript deals with a two-band model for KCuF_3 . The authors study by means of dynamical mean-field theory (DMFT) the out-of-equilibrium dynamics of such model perturbed by an electric field pulse or by a sudden photo-injection of holes or electrons in the eg conduction bands. They find a quasi-stationary transient state that has no counterpart in the equilibrium phase diagram, today referred to as hidden phases and observed, e.g., in 1T-TaS_2 . The calculations show that such a phenomenon arises by the different response to the perturbation of the spin with respect to the orbital degrees of freedom that characterise the model Hamiltonian. Specifically, while at equilibrium and upon increasing temperature, the magnetic order melts without affecting much the antiferro-orbital order, in the non-equilibrium dynamics the former seems to be more robust whereas the latter changes into a ferro-orbital order. This intriguing result suggests, as emphasised by the authors, novel directions where to search for hidden phases, namely materials where spin and orbital degrees of freedom are strongly entangled to each other, most notably transition-metal perovskites just like KCuF_3 or manganites.

I think that these results are quite interesting and convincing. However, I have some doubts about their relevance to real materials, even to KCuF_3 that is claimed to be well described by the model Hamiltonian in equations (1) and (2). Indeed, in this material as well as in the manganites, there is another important ingredient that is not included in the model Hamiltonian, i.e., the coupling to the lattice. For instance, it is known that the electron-only Hamiltonian cannot explain why the Jahn-Teller distortion in KCuF_3 persists up to at least 900K, see Physical Review B 87, 014109 (2013), and 96, 054107 (2017). I suspect that the electron-lattice coupling, strong in perovskites, will hinder the ferro-orbital transient state, though this is just my speculation.

I have a further comment. A hidden phase might be, for instance, a metastable phase whose energy and entropy are such that this phase is never stabilized at equilibrium. DMFT has the nice property of allowing accessing metastable phases by preventing dominant symmetry breakings. Therefore, I believe it may be possible that a magnetic ferro-orbital state is stabilized in an equilibrium-DMFT calculation by not allowing for antiferro-orbital order. This possibility, if confirmed, would shed light into the meaning of hidden phases and how to make them stable in out-of-equilibrium.

In conclusion, I have some doubts that the present version is indeed suitable for Nature Communications. The authors should address the role of the electron-lattice coupling, either including it in the simulation or justifying its neglect. Furthermore, I recommend the authors to ascertain whether the transient ferro-orbital state is actually a metastable state that can be uncovered at equilibrium by preventing different orbital orderings.

Reviewer #2:

Remarks to the Author:

The authors theoretically study (using non-equilibrium DMFT) the $\frac{3}{4}$ filled two-band Hubbard model to investigate the role the orbital degrees of freedom play in the switching to hidden states on femtosecond time scales in models with intertwined spin and orbital orders. The authors identify a mechanism governing the transition, based on the motion of charge following photoexcitation and the spatially anisotropic equilibrium state. They also find a hidden ordered state with a different combination of spin and orbital orders than observed in equilibrium. The authors propose that this could be detected in time-resolved X-ray scattering experiments via orbital order-related lattice

distortions.

The calculations are carried out for semi-realistic parameters for KCuF_3 at an equilibrium temperature of 100K. The authors consider three nonequilibrium protocols: (1) Excitation of the e_g system with an electric field pulse polarization parallel or (2) perpendicular to the ferromagnetically ordered planes, and (3) a sudden excitation of electrons into other orbitals through resonant excitations.

The authors find a slower and weaker melting of A-AFM order compared to AFO. They emphasize this is opposite of what is expected for a thermal melting: First spin order would weaken, then orbital order because of the smaller energy scale associated with the exchange interaction. In addition, the relaxation times for spin and orbital order are computed as a function of the excitation density, n_{ex} . The spin and orbital order decays approximately exponentially at long times.

The authors also compute the dependence of the pseudospin orbital order components X_1 and X_3 in the long-time limit. They find the rotation of this vector into opposite directions under increased equilibrium temperature (towards AFO) and photoexcitation strength (towards FO with small amplitude), showing non-equilibrium effects originating in the laser drive. The authors propose a mechanism based on a string of defects in the ordered background created by the motion of charge excitations: Due to the in-plane FM in the xy -plane the motion of charge excitations along the z -direction affects the spin order (little effect on spin while hopping in xy -plane) while orbital order is affected by hopping processes in all directions and would therefore lead to its more rapid decay. By contrast, thermal melting originates from spin-waves and orbital waves. The authors use a Kugel-Khomski model to explain the DMFT results for the polarization and time-dependence of the spin and orbital orders following photoexcitation.

Overall, the work is clearly presented and the results appear technically sound. The search for non-thermal effects in correlated electron systems is an important frontier in condensed matter physics research and this work adds an important example in a semi-realistic model. I therefore recommend the work for publication.

As a minor comment, I found the hopping process and the description of Figure 4 (b) a little dense and difficult to understand. I wonder if there is a way the authors can "expand" that figure into two components to explain a little more clearly the microscopic processes involved in the spin and orbital melting due to charge motion.

Referee 1:

The manuscript deals with a two-band model for KCuF_3 . The authors study by means of dynamical mean-field theory (DMFT) the out-of-equilibrium dynamics of such model perturbed by an electric field pulse or by a sudden photo-injection of holes or electrons in the eg conduction bands. They find a quasi-stationary transient state that has no counterpart in the equilibrium phase diagram, today referred to as hidden phases and observed, e.g., in 1T-TaS_2 . The calculations show that such a phenomenon arises by the different response to the perturbation of the spin with respect to the orbital degrees of freedom that characterise the model Hamiltonian. Specifically, while at equilibrium and upon increasing temperature, the magnetic order melts without affecting much the antiferro-orbital order, in the non-equilibrium dynamics the former seems to be more robust whereas the latter changes into a ferro-orbital order. This intriguing result suggests, as emphasised by the authors, novel directions where to search for hidden phases, namely materials where spin and orbital degrees of freedom are strongly entangled to each other, most notably transition-metal perovskites just like KCuF_3 or manganites.

I think that these results are quite interesting and convincing. However, I have some doubts about their relevance to real materials, even to KCuF_3 that is claimed to be well described by the model Hamiltonian in equations (1) and (2). Indeed, in this material as well as in the manganites, there is another important ingredient that is not included in the model Hamiltonian, i.e., the coupling to the lattice. For instance, it is known that the electron-only Hamiltonian cannot explain why the Jahn-Teller distortion in KCuF_3 persists up to at least 900K, see Physical Review B 87, 014109 (2013), and 96, 054107 (2017). I suspect that the electron-lattice coupling, strong in perovskites, will hinder the ferro-orbital transient state, though this is just my speculation.

REPLY:

We thank the referee for the insightful comments and suggestions! We agree that the Jahn-Teller effect is an important mechanism in the cooperative electron-lattice dynamics of realistic perovskites, especially in the long-time limit. Nevertheless, we believe that the main conclusion of our manuscript is still relevant to real materials. The reasons are as follows:

1) Even though the Jahn-Teller coupling will affect the laser-excited dynamics of KCuF_3 , the electronic mechanism discussed in the manuscript should remain a critical ingredient of the scenario. In the real material, the dynamics of the orbital order will be influenced cooperatively both by lattice and electronic effects, but nevertheless, the electronic contribution to the orbital exchange remains an essential contribution. In fact, experiments suggest that the lattice dynamics subsequent to photo-excitation might actually be driven by the electronic excitation, such as in [Ref. 14]. Moreover, one essential aspect of our mechanism is the fact that the non-thermal melting of spin and orbital order happens at a different pace as compared to the thermal melting. We argue below that this

should remain true even when the electron-lattice coupling is taken into account:

In general, the electronic mechanism favors weaker AFO than A-AFM, while the Jahn-Teller effect tends to stabilize the AFO. The interplay of the two effects depends on the relative coupling strength. However, both in the limits of strong and weak electron-lattice coupling, the general conclusion of weaker melting of A-AFM than AFO should not be dramatically modified:

- For weak enough electron-phonon coupling, this is true by construction. The time scale of the charge-relaxation and the non-thermal melting of orders can be much shorter than that of the Jahn-Teller modes (greater than 100 fs, as in [Ref. 14]), and thus the electronic mechanism is robust at least on the ultrashort time scale, and will provide the driving force for a subsequent combined electron lattice dynamics.
- In the limit of strong electron-lattice coupling, one can no longer view the problem as electronic dynamics in a frozen lattice environment. Instead, the photo-induced carriers will be dressed by Jahn-Teller modes and acquire a strong polaronic nature. One can speculate that the hopping of these polarons would be similar to the hopping of the electrons in our study, though with renormalized parameters, such that the non-thermal melting is analogous as discussed in our work. The actual investigation of this limit, though numerically much harder and thus beyond the scope of the present publication, may be possible using similar methods as introduced in [Phys. Rev. B 88, 165108], which is an interesting direction for future research.

Hence, in the two limits, the electronic mechanism for the non-thermal melting should allow relevant insights into the realistic scenario. The regime of intermediate Jahn-teller coupling can be more complicated, while studying the competition of the two effects is well beyond the scope of the manuscript and requires the development of convenient numerical tools.

In summary, even when the Jahn-Teller effect is considered, the electronic mechanism is still a crucial driving force in the full electron-lattice dynamics. Our results have identified this non-trivial mechanism and laid a solid foundation for future studies, e.g. of the competition between various effects in specific materials.

2) Secondly, we stress that the manuscript is not intended to be an *ab initio* description of the photo-induced dynamics in KCuF_3 . On the contrary, with the model simulation, we have demonstrated that a pure electronic mechanism can already lead to hidden phases without the assistance of other effects such as Jahn-Teller coupling. Considering the scarcity of microscopic theories of hidden phases in multi-orbital strongly correlated materials, the identification of a pure electronic mechanism is certainly a major step and will provide valuable intuition for further studies. Moreover, the mechanism, where charge excitations transfer energy to different orders at different rates, is conceptually transpar-

ent and can readily be generalized to other situations with different spin-orbital ordering. In particular, we have chosen the e_g -orbital Hubbard model to demonstrate the general idea since it is computationally easier to solve. But the general mechanism should also apply to materials with active t_{2g} orbitals, weak electron-phonon interactions, and even spin-orbit interactions.

To clarify these points, we have cited the two papers mentioned by the referee ([40, 41]) and added the following sentences in the first paragraph of the Discussion section:

... subsequent dynamics of non-thermal order parameters [14]. In particular, the Jahn-Teller effect can be nonnegligible in realistic perovskites [40,41], but the electronic mechanism would still be a key driving force, among other effects, of the full electron-lattice dynamics. One possible scenario is suggested by recent experiments [7, 14], where the subsequent lattice dynamics is driven by the fast change in electronic degrees of freedom. This should be the case when the electron-lattice coupling is weak enough and only affects the dynamics on longer time scales than the electronic processes. In the strong coupling limit, on the other hand, the electrons can be dressed with lattice distortions to form polarons. With renormalized parameters, the electronic mechanism of weaker and slower melting of AFO than A-AFM could still be established in the polaron dynamics. Thus, in both cases, the opposite rotation of the orbital pseudospin in the equilibrium and non-equilibrium regimes might reveal itself by inducing opposite Jahn-Teller-like distortions through electron-phonon coupling, which can possibly be detected by time-resolved X-ray diffraction techniques. The competition between Jahn-Teller effect and the electronic mechanism in the intermediate coupling regime can be more complicated and requires further studies.

Referee 1:

I have a further comment. A hidden phase might be, for instance, a metastable phase whose energy and entropy are such that this phase is never stabilized at equilibrium. DMFT has the nice property of allowing accessing metastable phases by preventing dominant symmetry breakings. Therefore, I believe it may be possible that a magnetic ferro-orbital state is stabilized in an equilibrium-DMFT calculation by not allowing for antiferro-orbital order. This possibility, if confirmed, would shed light into the meaning of hidden phases and how to make them stable in out-of-equilibrium.

REPLY:

Indeed, the possible metastability of the hidden phase is a very interesting question.

First of all, we should clarify the different meanings of a “metastable state” in different contexts:

- 1) A metastable equilibrium state whose free energy is higher than the true

ground state. In this case, one would simply have multiple possible DMFT solutions, which depend on the initial guess of the DMFT iteration.

- 2) A DMFT solution obtained by constraining the type of orbital ordering.
- 3) A long-lived non-equilibrium steady-state with charge excitations.

Concerning the second notion of metastability, one can indeed obtain converged ferro-orbital solutions by constraining the possible orbital ordering, but mean-field calculations within the Kugel-Khomskii model suggest that these solutions are thermodynamically unstable when the constraint is released. More importantly, the observed hidden phase is essentially different from a metastable equilibrium DMFT solution of type 1) and 2), because it does not satisfy the fluctuation-dissipation theorem: The presence of charge excitations gives rise to a non-Fermi-Dirac distribution in the single-particle spectrum, which is in sharp contrast to a true equilibrium state (Fig. 4a).

Hence, we believe it is crucial to impose a non-thermal charge distribution as shown in Fig. 4a (which is more like a separate Fermi-Dirac distribution for electron and hole-like charges) to discuss the stability of the hidden phase and its slow evolution together with the lattice dynamics on the picosecond timescale. Such a non-thermal distribution may be implemented in a steady-state formalism in the future. We have added comments about this point in the second paragraph of Discussion section:

... and thereby further control the dynamics on the ps timescale. It is worth noting that, although one can explicitly impose the ferro-orbital ordering in a constrained DMFT simulation, the obtained solution is in general unstable in equilibrium. Hence, the presence of charge excitations should play a critical role in stabilizing the hidden phase. To investigate those possibilities in detail...

Referee 1:

In conclusion, I have some doubts that the present version is indeed suitable for Nature Communications. The authors should address the role of the electron-lattice coupling, either including it in the simulation or justifying its neglect. Furthermore, I recommend the authors to ascertain whether the transient ferro-orbital state is actually a metastable state that can be uncovered at equilibrium by preventing different orbital orderings.

REPLY:

We once again thank the referee for the valuable comments. We hope that with the clarifications above, the revised manuscript is now suitable for Nature Communications.

Referee 2:

The authors theoretically study (using non-equilibrium DMFT) the 3/4 filled

two-band Hubbard model to investigate the role the orbital degrees of freedom play in the switching to hidden states on femtosecond time scales in models with intertwined spin and orbital orders. The authors identify a mechanism governing the transition, based on the motion of charge following photoexcitation and the spatially anisotropic equilibrium state. They also find a hidden ordered state with a different combination of spin and orbital orders than observed in equilibrium. The authors propose that this could be detected in time-resolved X-ray scattering experiments via orbital order-related lattice distortions.

The calculations are carried out for semi-realistic parameters for KCuF_3 at an equilibrium temperature of 100K. The authors consider three nonequilibrium protocols: (1) Excitation of the eg system with an electric field pulse polarization parallel or (2) perpendicular to the ferromagnetically ordered planes, and (3) a sudden excitation of electrons into other orbitals through resonant excitations.

The authors find a slower and weaker melting of A-AFM order compared to AFO. They emphasize this is opposite of what is expected for a thermal melting: First spin order would weaken, then orbital order because of the smaller energy scale associated with the exchange interaction. In addition, the relaxation times for spin and orbital order are computed as a function of the excitation density, n_{ex} . The spin and orbital order decays approximately exponentially at long times.

The authors also compute the dependence of the pseudospin orbital order components X_1 and X_3 in the long-time limit. They find the rotation of this vector into opposite directions under increased equilibrium temperature (towards AFO) and photoexcitation strength (towards FO with small amplitude), showing non-equilibrium effects originating in the laser drive. The authors propose a mechanism based on a string of defects in the ordered background created by the motion of charge excitations: Due to the in-plane FM in the xy -plane the motion of charge excitations along the z -direction affects the spin order (little effect on spin while hopping in xy -plane) while orbital order is affected by hopping processes in all directions and would therefore lead to its more rapid decay. By contrast, thermal melting originates from spin-waves and orbital waves. The authors use a Kugel-Khomski model to explain the DMFT results for the polarization and time-dependence of the spin and orbital orders following photoexcitation.

Overall, the work is clearly presented and the results appear technically sound. The search for non-thermal effects in correlated electron systems is an important frontier in condensed matter physics research and this work adds an important example in a semi-realistic model. I therefore recommend the work for publication.

As a minor comment, I found the hopping process and the description of Figure 4 (b) a little dense and difficult to understand. I wonder if there is a way the authors can expand that figure into two components to explain a little more clearly the microscopic processes involved in the spin and orbital melting due

to charge motion.

REPLY:

We sincerely thank the referee for the appreciation of our work and the suggestion. We have modified the figure to demonstrate the processes more clearly. The color of the fully occupied lattice site (created by non-equilibrium excitations) is changed to orange, which is distinguished from other sites. The caption is changed accordingly.

(a) Relaxation of charge excitations and the photo-excited state in the long time limit. $G^<(\omega, t)$ at $\omega > 0$ corresponds to the distribution of 4-particle excitations at $\omega > 0$ while $G^>(\omega, t)$ at $\omega < 0$ shows the distribution of doublons (see methods). The red solid line is the density of states (spectral function) at $t = 50$. (b) The charge excitations created by pulses perpendicular/parallel to the ferromagnetic plane. A “parallel pulse” ($E \parallel x, y$) creates spin-triplet excitations in process 4, while a “perpendicular pulse” ($E \parallel z$) is polarized along the AF-axis and directly creates spin-singlet excitations in process 3. Processes 1 and 2 indicate possible hopping processes of 4-particle excitations. **The lattice site of the 4-particle excitation is of orange color.** Blue arrows indicate the polarization of the electric field. (c) The non-equilibrium melting of spin and orbital order under a one-cycle electric field pulse of $E_0 = 0.6$. The solid (dashed) lines show X_3 (S_z) normalized by their initial values.

Reviewers' Comments:

Reviewer #1:

Remarks to the Author:

I am satisfied by the authors' reply and by the revisions of the manuscript, and therefore I am now favourable to its publication.